# Joint Diffusion for Universal Hand-Object Grasp Generation

**Jinkun Cao***  *jinkunc@andrew.cmu.edu*
*Carnegie Mellon University*

**Jingyuan Liu**  *jingyliu@adobe.com*
*Adobe*

**Kris Kitani**  *kmkitani@andrew.cmu.edu*
*Carnegie Mellon University*

**Yi Zhou***  *yizhou@roblox.com*
*Roblox*

**Reviewed on OpenReview:** *https://openreview.net/forum?id=TZOztsYR6x*

## Abstract

Predicting and generating human hand grasp over objects is critical for animation and robotic tasks. In this work, we focus on generating both the hand and objects in a grasp by a single diffusion model. Our proposed Joint Hand-Object Diffusion (JHOD) models the hand and object in a unified latent representation. It uses the hand-object grasping data to learn to accommodate hand and object to form plausible grasps. Also, to enforce the generalizability over diverse object shapes, it leverages large-scale object datasets to learn an inclusive object latent embedding. With or without a given object as an optional condition, the diffusion model can generate grasps unconditionally or conditional to the object. Compared to the usual practice of learning object-conditioned grasp generation from only hand-object grasp data, our method benefits from more diverse object data used for training to handle grasp generation more universally. According to both qualitative and quantitative experiments, both conditional and unconditional generation of hand grasp achieves good visual plausibility and diversity. With the extra inclusiveness of object representation learned from large-scale object datasets, the proposed method generalizes well to unseen object shapes.

## 1 Introduction

In this paper, we explore the generation of hand-object grasps. Unlike traditional approaches that fit to a limited set of objects found in 3D hand-object interaction datasets, our goal is to generalize across diverse object geometries. We develop a joint diffusion model capable of generating hand poses either conditionally on or jointly with object shapes. Our method approaches objects from a purely geometric perspective, eliminating the need for language descriptions or category labels. Despite having access to limited hand-object interaction datasets, our model learns a comprehensive object shape embedding by leveraging large-scale object shape datasets. It generates paired hand and object grasps by denoising a joint latent representation.

To generate hand grasp over objects, existing works (Taheri et al., 2020; Lu et al., 2023; Fan et al., 2023) typically rely on datasets with *full-stack* 3D annotations (Taheri et al., 2020; Fan et al., 2023; Yang et al., 2022) from object scans and hand pose capture. However, with the difficulty of capturing 3D object models and hand gestures, this area of research faces notorious limitations of annotated data, e.g., only dozens of

---

*: Jinkun and Yi were affiliated with Adobe Research when the main part of the work was conducted

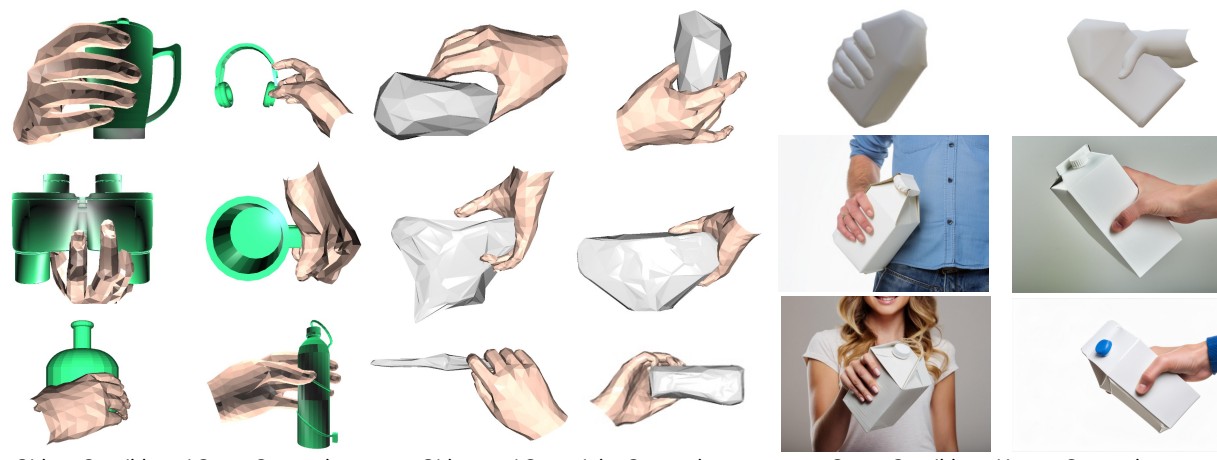

Figure 1: Applications of our proposed method Joint Hand-Object Diffusion (JHOD). Left: generating the hand grasp on an unseen object. Middle: jointly sampling a combination of the hand and the object for a plausible grasp. Right: the generated grasp can be used as guidance for generating photo-realistic grasping images using existing image generation tools (Adobe, 2024).

object shapes are available in a dataset (Fan et al., 2023; Taheri et al., 2020). Moreover, existing datasets are designed for different hand parametric models (Miller & Allen, 2004; Jian et al., 2023; sha; Wang et al., 2023; Romero et al., 2022; Taheri et al., 2020; Yang et al., 2022), which prevents directly combining different data resources to scale up. The limited scale of object annotations causes the method to overfit to biased object shapes and thus bad generalizability of generating grasps on unseen objects.

Humans have the remarkable ability to transfer grasping patterns to different objects and make plausible grasping choices for unseen objects based solely on their geometry. While prior knowledge, such as object category or functionality, can aid in manipulation, it is often not essential for determining a grasp.

We refer to this capability as *universal grasp generation*. We argue that the key to universally generating hand-object grasps lies in developing an inclusive object shape embedding. This is unattainable if we train object embeddings using datasets with limited quantity and diversity. Although comprehensive 3D annotations are scarce, we have access to large-scale 3D object datasets. In this work, we investigate whether these datasets can be leveraged to enhance hand-object grasp generation by jointly modeling hand and object representations in a latent space. Our model is designed to accommodate *partial supervision* during training, accepting available data such as hand poses or object shapes. This flexibility significantly broadens the range of datasets that can be utilized for learning. Our model can generate complete 3D grasping scenes, including both the posed hand and object, or a posed hand conditioned on a given object geometry. We believe that this flexible approach to generating hand-object interactions, without relying on domain-specific auxiliary information, represents a significant step toward human-like universal grasp generation.

Following such an intention, we propose our Joint Hand-Object Diffusion (JHOD) model. It follows the latent diffusion model (Rombach et al., 2022) (LDM) to encode and ensemble different modalities into a latent space and then generate plausible samples by the probabilistic denoising process (Ho et al., 2020). To learn from more diverse objects, we construct the model by leveraging the object shape encoding and decoding networks pre-trained on the large-scale generic 3D object shape dataset (Chang et al., 2015). As we aim to derive a universal grasp generation solution, we remove the category-specific information when constructing the latent code for objects and encode the object shapes in a purely geometric fashion.

As the hand articulation and positions are always coupled to the object shape in hand-object grasping, we design the latent diffusion to denoise the latent code from different modalities as a whole. We decouple each modality representation to allow model training by combining data from different resources with heterogeneous annotations. Therefore, our method has independent encoder and decoder networks for different

modalities and can optimize the generation of each modality solely. Such a design relieves the limitation of object diversity in hand-object interaction datasets as we could leverage the data from generic object shape datasets to tune the object generation part. The model is trained to generate the hand part regarding the corresponding object shape embedding, so an inclusive object generation improves the corresponding hand grasp generation on diverse object shapes. To achieve disentangled modality representations to boost training, we propose to use asynchronous denoising schedulers for different modalities during noise diffusion and denoising in training.

In conclusion, we have developed a joint hand-object grasp generation model capable of sampling from either the joint distribution of both hand and object or the object-conditioned distribution for hand generation. This model effectively leverages training resources with varying annotations. Through both qualitative and quantitative evaluations, our method demonstrates strong performance in grasp generation for both unconditional and object-conditioned scenarios. Thanks to the posterior generation learned from extensive data resources, our model significantly outperforms others in generating grasps for out-of-domain object shapes, a crucial aspect of universal grasp generation. Given that no existing works have supported the end-to-end generation of both hand and object to form a grasp, we believe our approach is pioneering in the field of universal grasp generation.

With the flexibility to generate both modalities in hand grasp, our model can facilitate many downstream applications. The grasps are good references for generating photo-realistic images with good geometry alignment and significantly fewer artifacts. We first generate grasps in 3D and use them as conditional signals for image generation (Adobe, 2024) can help produce high-quality hand images with challenging gestures and view angles. We demonstrate using grasp from our model as the condition improves grasp alignment on images and relieves artifacts. Some examples of object-conditioned grasp generation, joint generation, and image generation with grasp rendering are presented in Figure 1. The main contribution of this work is to propose a generative model capable of generating hand grasp in either unconditional or object-conditioned fashions and the corresponding strategy to enhance the performance with limited available 3D hand-object grasp annotations.

## 2 Related Works

**Hand Grasp Generation.** Modern generative models (Ho et al., 2020) are recently introduced into hand grasp reconstruction (Ye et al., 2023a) and generation. Hand grasp reconstruction asks for hand grasp rendering aligned with images while hand grasp generation asks for hand grasp visually or physically plausible. GrabNet (Taheri et al., 2020) is a commonly used method for hand grasp generation but it can hardly be generalized to diverse object shapes. HOIDiffusion (Zhang et al., 2024b) uses the 3D hand grasp from GrabNet and an image diffusion model to generate hand-object interaction images. AffordanceDiffusion (Ye et al., 2023b) also studies the generation of hand-object interaction for image synthesis. Instead of generating hand-object interaction images or videos, we focus on improving hand-object interaction in 3D space. Many recent works are based on diffusion models (Ye et al., 2024; Lu et al., 2023; Christen et al., 2024) or physics-based simulators (Zhang et al., 2024a; Luo et al., 2024a; Xu et al., 2023; Luo et al., 2024b). Compared to existing works which typically require a given object (Taheri et al., 2020; Zhang et al., 2024b; Ye et al., 2023b; Zhou et al., 2024) shape as input, we desire generating a hand grasp over a given object or generating both the object and hand to form a plausible grasp.

**Multi-Modal Generation.** With the rise of diffusion models, multi-modal generation has been extended in many areas, such as image+text generation (Bao et al., 2023) and audio+pixel generation (Ruan et al., 2023). Our model can jointly generate the hand and object in a grasp, making another type of multi-modal generation. A concurrent work UGG (Lu et al., 2023) studies to generate both hand and object to form a grasp but it uses the ShadowHand parametric model to leverage the large-scale synthetic ShadowHand grasping datasets (Wang et al., 2023) and the final results are optimization-based instead of directly from the generative model. On the other hand, another concurrent work G-HOP (Ye et al., 2024) builds a multi-modal hand-object grasp prior by encoding object shape and hand poses into a unified latent space but a language prompt is required to provide necessary conditional information. In this work, we focus on a dual-modal latent diffusion model to generate the modalities of a 3D object and hand at the same time

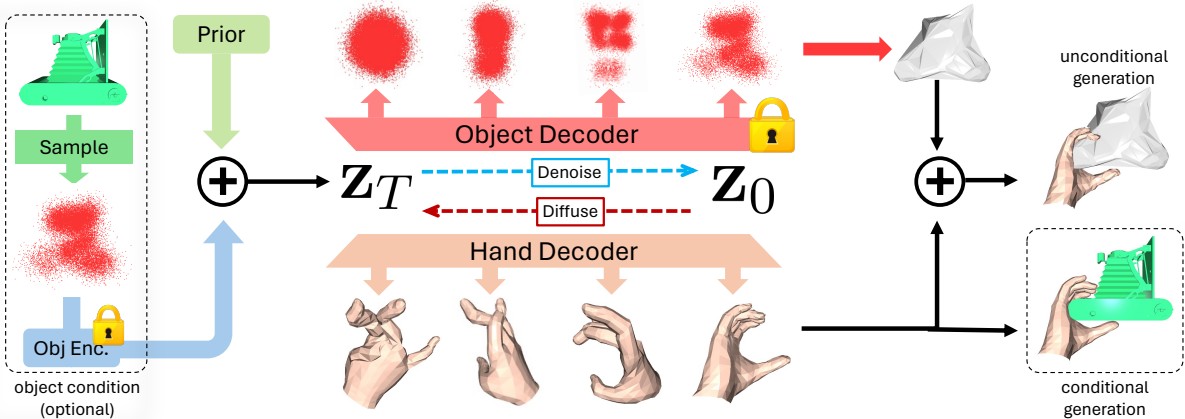

Figure 2: The illustration of our proposed Joint Hand-Object Diffusion (JHOD). We present the main modules of the model while removing the secondary modules for simplicity. The optional object condition can turn the generation into object-conditioned.

without requiring optimization or auxiliary conditions. The hand grasp is purely geometric-based. It allows more flexibility for downstream tasks, such as generating photo-realistic images by using the rendering of generated 3D hand-object interaction as a condition.

## 3 Method

In Section 3.1, we first provide the formal problem formulation. We then review the related preliminary knowledge for methodology and data representation in Section 3.2. Finally, we introduced our proposed method in Section 3.3.

### 3.1 Problem Formulation

An instance of hand-object grasp consists of a posed hand, denoted as $\mathbf{H}$, and a posed object, denoted as $\mathbf{O}$. To generate a hand-object grasp unconditionally, we model the joint distribution of hand and object:

$$\{\mathbf{O}, \mathbf{H}\} \sim \mathcal{D}_\Phi. \tag{1}$$

Also, in the community of visual perception, animation, and robotics, people are interested in generating hand grasp conditioned on certain objects, which can be formulated as

$$\{\mathbf{H}\} \sim \mathcal{D}_\Phi|_{\mathbf{O}_g}, \tag{2}$$

where $\mathbf{O}_g$ is a provided object shape. In the previous works, the unconditional generation has been under-explored. The methods for conditional hand grasp generation are usually required with no flexibility for object shape generation. They suffer from bad robustness to different object shapes. This is because the existing datasets with 3D annotations of both modalities are notoriously limited in covered object shapes due to the expansiveness of scanning the object shape and capturing the corresponding hand poses. These methods' limitation is underestimated because the object shapes used in training and testing in existing datasets share similar scales and geometries. In this work, we aim to solve both unconditional and object-conditioned grasp generation by a single model. We also wish to enhance the robustness of object shapes by deriving a more inclusive object embedding which also improves the generalizability of the hand part generation.

### 3.2 Preliminaries

#### 3.2.1 Latent Diffusion

Our method follows the latent diffusion models (LDM) (Rombach et al., 2022; Sohl-Dickstein et al., 2015) to generate samples in two modality spaces, i.e., articulable hands, and rigid objects. LDM diffuses and denoises in a latent space instead of on the raw data representations.

To diffuse a data sample, given a clean latent code $z_0$ from a data sample $y \sim \mathcal{D}_y$, i.e., $z_0 = \mathcal{E}(y) \sim \mathcal{D}_z$, we add noise following a Markov noise process:

$$q(z_t|z_{t-1}) = \mathcal{N}(\sqrt{\alpha_t}z_{t-1}, (1-\alpha_t)I), \quad t \in [1, T], \tag{3}$$

where $\alpha_t \in (0, 1)$ are constant hyper-parameters determined by the time step $t$ and a noise scheduler. Though the typical diffusion model (Ho et al., 2020) trains a denoiser to predict the noise added per time step during the diffusion stage, there is another line of diffusion models (Ramesh et al., 2022; Tevet et al., 2022) that learns to recover the clean data sample directly. The loss turns to

$$L_{\text{LDM}} := \mathbb{E}_{z_0 \sim \mathcal{D}_z, t \sim [1,T]}\Big[||z_0 - G_\Phi(z_t, t)||_2^2\Big], \tag{4}$$

requiring a generative model $G_\Phi$ to predict a clean latent code conditioned to a noisy latent code on a corresponding time step. We follow this paradigm of training diffusion models in this work. Besides the diffusion model itself, we also need the conversion between the raw data representation and the latent code. After recovering a clean code $\hat{z}_0$, we can convert it to the raw data representation by a decoder network $\mathcal{X}$ which is trained to be the inverse of $\mathcal{E}$, i.e., ideally $\mathcal{X}(z_0) \sim \mathcal{D}_y$. Therefore, there are three modules with learnable weights: an encoder network $\mathcal{E}$, a decoder network $\mathcal{X}$, and the denoiser network $G_\Phi$.

#### 3.2.2 Modality Representations

Here we introduce the raw representations of the modalities involved in interrupting a hand-object interaction.

**Hand.** We follow the MANO (Romero et al., 2022) parametric model to represent hands. Compared to another parametric model ShadowHand (sha), MANO is designed for animating non-rigid real human hands instead of robotic hands thus a better fit for animation and photo-realistic generalization tasks. However, its higher complexity causes difficulty in grasping data synthesis. Therefore, there is no MANO-based hand grasping dataset as large-scale as the synthetic ShadowHand-based datasets (Wang et al., 2023) yet. This prevents research on the joint generation of MANO hands and objects in a straightforward supervised learning fashion. In this work, we directly use MANO parameters as joint rotation angles, i.e. $\theta \in \mathbb{R}^{48}$, and the translation of hands $\mathbf{t}_h \in \mathbb{R}^3$. We use constant mean shape parameters $\beta_0 = \mathbf{0}^{10}$ during linear blend skinning. The parameters of hands are concatenated and encoded by a hand encoder $\mathcal{E}_H$ to be the hand latent code $\mathbf{y}_H \in \mathbb{R}^{51}$.

**Object.** For object representation, we use the point cloud latent code as $\mathbf{y}_O = \mathbf{h} \in \mathbb{R}^{N \times 4}$, where $N$ is the number of points. As the grasp is modeled in the object-centric frame, no translation or orientation is required in the object pose and shape representation. This convention is aligned with the definition of data in many hand-object grasp datasets, such as OakInk (Yang et al., 2022).

#### 3.2.3 Object Generation

The limited amount and high category-specific bias of the objects in hand-object interaction (HOI) datasets prevent learning generating grasps robust and generalizable to diverse object shapes. When training on these datasets only, the learned object embedding is usually biased and overfit. By leveraging the large-scale object-only 3D datasets, we can derive a more robust object encoding for the grasp generation than learning from the HOI datasets solely. For the object encoding part, we borrow the point-cloud-based object shape generation method LION (Vahdat et al., 2022). Existing HOI generation methods can only learn object shapes from HOI datasets, including only dozens of objects, while LION learns from a much larger basis, i.e., more than 50,000 objects in ShapeNet (Chang et al., 2015). LION generates objects in two stages. In the

first step, it derives a global latent code $\mathbf{z}_0^G \in \mathbb{R}^{D_G}$ from a posterior distribution $q_\phi(\mathbf{z}_0^G|\mathbf{P})$ where $\mathbf{P} \in \mathbb{R}^{u \times 3}$ is the object point cloud. Then it samples a point cloud latent $\mathbf{h}_0$ from a posterior $q_{\mathcal{E}_O}(\mathbf{h}_0|\mathbf{P}, \mathbf{z}_0^G)$. LION generates the object shape by a reverse process from sampled latent code

$$p_{\mathcal{E}_O, \psi, \gamma}(\mathbf{P}, \mathbf{h}_0, \mathbf{z}_0^G) = p_{\mathcal{X}_O}(\mathbf{P}|\mathbf{h}_0, \mathbf{z}_0^G)p_\psi(\mathbf{h}_0|\mathbf{z}_0^G)p_\gamma(\mathbf{z}_0^G), \tag{5}$$

where $\mathcal{X}_O$ is the decoder network, $\psi$ is the generator of point cloud latent code and $\gamma$ is the distribution of global latent code. In this work, we remove the global latent code from LION to consider the object shape from a purely geometric perspective. We encode and decode the object between point clouds and latent code from the point cloud latent $\mathbf{h}_0$ only. We convert the object generation part to

$$p_{\mathcal{E}_O, \psi'}(\mathbf{P}, \mathbf{h}_0) = p_{\mathcal{X}_O}(\mathbf{P}|\mathbf{h}_0)p_{\psi'}(\mathbf{h}_0), \tag{6}$$

where $\psi'$ is fine-tuned from $\psi$ by replacing the global prior $\gamma$ with a standard Gaussian. Thanks to the pre-training from large-scale object datasets (Chang et al., 2015), we could use the weights of $\mathcal{E}_O$ and $\mathcal{X}_O$ from LION directly. By freezing the encoder and decoder, we fine-tune the object prior distribution with the object shapes from HOI datasets.

### 3.3 Joint Hand-Object Diffusion (JHOD)

We now introduce our proposed Joint Hand-Object Diffusion (JHOD). There are two main goals. First, we would like to use the training data with heterogeneous annotations so that we can go beyond the limited object shapes in HOI datasets. Second, we want a single model capable of object-conditioned hand grasp generation and unconditional hand-object joint generation. We present the overall illustration and key designs of our proposed method in Figure 2.

#### 3.3.1 Latent Codes for Different Modalities

For each modality, we have an MLP-based encoding network to convert the raw representation into latent codes:

$$\mathbf{z}_0^H = \mathcal{E}_H(\mathbf{y}_H), \quad \mathbf{z}_0^O = \mathcal{E}_O(\mathbf{y}_O), \tag{7}$$

with the dimensions $\mathbf{z}_0^H \in \mathbb{R}^{1 \times 512}$ amd $\mathbf{z}_0^O \in \mathbb{R}^{4 \times 512}$ . They are concatenated to be the final latent code for the whole hand-object interaction configuration

$$\mathbf{z}_0 = \mathbf{z}_0^H \oplus \mathbf{z}_0^O \in \mathbb{R}^{5 \times 512}. \tag{8}$$

This design allows us to jointly process different modalities without unnecessary entanglement. So we can train the model with partial modality supervision without corrupting the parameters for other modalities.

#### 3.3.2 Asynchronous Denoising Schedulers

We propose to generate either both hand and object or only the hand given an object shape. To realize this, we need a certain degree of entanglement between the two modalities as we need them to cooperate to form a valid grasp. However, we still desire certain disentanglement so that we could supervise the training with partially annotated data, e.g., 3D object shape data. In the usual fashion of organizing multi-modal latent codes in latent diffusion models, the latent codes of different modalities always have the same scale of noise corruption as they share the noise scheduler. In contrast, we desire that the two modalities can be denoised with respect to an arbitrary level of noise in the other modality. Therefore, we propose to use asynchronous denoising schedulers for this purpose during training. Instead of using a single noise scheduler, we have two schedulers $t_H$ and $t_O$ to control the noise patterns in the hand latent code and the object latent code respectively. Following Equation (3) we derive the corrupted latent codes as

$$\mathbf{z}_{t_H}^H = \Pi_{t=1}^{t_H}q(\mathbf{z}_t^H|\mathbf{z}_{t-1}^H, \mathbf{z}_{t-1}^O), \quad \mathbf{z}_{t_O}^O = \Pi_{t=1}^{t_O}q(\mathbf{z}_t^O|\mathbf{z}_{t-1}^O), \tag{9}$$

with the same time step range $t_O, t_H \in [1, T]$. This design allows the diffusion to deal with the noise in each modality separately. During training, it allows using object-only data to supervise the object part only. During sampling, it allows the model to generate only the hand given an object shape as the condition. Similar to bao2023unidiffuser, we will learn the joint distribution for hand-object pairs and the marginal distribution for the hand at the same time in an end-to-end fashion by manipulating the time schedulers.

### 3.3.3 Training and Sampling

**Training.** When training on HOI data samples, with the asynchronous noise schedulers, we can derive the assembly of two corrupted latent codes as

$$\mathbf{z}_{t_H, t_O} = \mathbf{z}_{t_H}^H \oplus \mathbf{z}_{t_O}^O. \tag{10}$$

Due to the element-wise property of MSE loss, we can calculate and back-propagate the gradient to a certain part of the latent codes independently. We leverage this property to train the unconditional and conditional generation at the same time. To learn the joint distribution, we apply the unconditional generation loss

$$L_{\text{uncond.}} = \mathbb{E}_{z_0 \sim \mathcal{D}_{HOI}, \{t_H, t_O\} \sim [1, T]} \left[ ||z_0 - G_\Phi(z_{t_H, t_O}, t_H, t_O)||_2^2 \right], \tag{11}$$

where $\mathcal{D}_{HOI}$ is the distribution of the HOI datasets (Taheri et al., 2020; Yang et al., 2022). We also train the model to learn the object-conditioned grasp generation. Given an object latent code $\mathbf{z}_0^O$, we set $t_O = 0$ indicating that the object part is noise-free and serves as a condition to derive the marginal distribution. The object-conditioned generation loss is thus

$$L_{\text{cond.}} = \mathbb{E}_{z_0 \sim \mathcal{D}_{HOI}, t_H \sim [1, T]} \left[ ||z_0 - G_\Phi(z_{t_H}, t_H)||_2^2 \right], \tag{12}$$

with $z_{t_H} = z_{t_H}^H \oplus \mathbf{z}_0^O$. The conditional and unconditional generation losses require uncorrupted and corrupted object latent codes respectively. In practice, we combine these two training objectives in a 1:1 ratio for a single draw of training data batch. This training strategy is similar to the classifier-free guidance (Ho & Salimans, 2022) for diffusion model training. However, the optional condition (object) exists in the input and output instead of in an independent condition vector.

Finally, we could leverage the large-scale 3D object shape datasets to train the object distribution only. This learning goal is designed to improve the object latent code by exposing it to a broader distribution of object shapes. We select object samples from both HOI datasets $\mathcal{D}_{HOI}$ and object-only datasets $\mathcal{D}_O$. The loss turns to

$$L_{\text{obj.}} = \mathbb{E}_{z_0^O \sim \mathcal{D}_O \cup \mathcal{D}_{HOI}, t_O \sim [1, T]} \left[ ||z_0 - G_\Phi(z_{t_O}, t_O)||_2^2 \right], \tag{13}$$

where we apply no supervision on the hand branch. While a hand latent code is present as a part of the joint latent code, it is randomly initialized and remains untrained. The latent codes only take the object part as the variable:

$$\mathbf{z}_0 = \epsilon \oplus \mathbf{z}_0^O, \quad \mathbf{z}_{t_O} = \epsilon \oplus \mathbf{z}_{t_O}^O, \quad s.t. \quad \epsilon \sim \mathcal{N}(0, I). \tag{14}$$

Obviously, the gradient only influences the object-related parts while keeping the hand part as-is. During training, the object encoder and decoder networks are pre-trained on the ShapeNet (Chang et al., 2015) and frozen. All other encoder, decoder, and denoiser networks are trained end to end.

Similar to the proximity sensor features (Taheri et al., 2024) and the Grasping Filed (Karunratanakul et al., 2020), we use a distance field to help capture the relation between the hand and the object. Since we eliminate the requirement of object templates to accommodate more general datasets, we can not define the field using object keypoints (Fan et al., 2023) or templated vertices. Instead, we define the distance field as the vectors between each hand joint and the 10 closest points in the object points. Therefore, we have the distance field as $\mathbf{y}_f \in \mathbb{R}^{16 \times 30}$. When the data sample is drawn from HOI datasets, we would also supervise the distance field calculated from the recovered hand gesture and object shape:

$$L_{\text{distance}} = ||\mathbf{y}_f - \hat{\mathbf{y}}_f||_2^2, \tag{15}$$

Besides supervision of the raw MANO parameters for hand, we also supervise the joint position by the loss

$$L_{\text{hand\_xyz}} = ||\text{LBS}(\mathbf{y}_H) - \text{LBS}(\hat{\mathbf{y}}_H)||_2^2, \tag{16}$$

where $\text{LBS}(\cdot)$ is the linear blend skinning process by MANO parametric model to derive the joint kinematic positions. With all the losses introduced, we could train the diffusion model using data with heterogeneous annotations and for unconditional and conditional generation at the same time. The overall loss is

$$L = \mathbb{1}_{z_0 \sim D_{HOI}}(L_{\text{uncond.}} + L_{\text{cond.}} + L_{\text{distance}} + L_{\text{hand\_xyz}}) + L_{\text{obj.}}. \tag{17}$$

Table 1: Quantitative evaluation of grasp generation. All models are trained using GRAB and OakInk-shape training set. We evaluate the methods on the objects from OakInk-test set, objects generated by JHOD, and objects from the ARCTIC dataset.

| Objects | Methods | FID ↓ | Pene. Dep. ↓ | Intsec. Vol. ↓ | Sim. Disp. Mean ↓ | User Scor.↑ |
|---|---|---|---|---|---|---|
| OakInk-test | GrabNet | 26.96 | 0.70 | 11.93 | 3.14 | 3.80 |
| | GrabNet-Refine | 25.26 | 0.63 | 4.44 | 2.78 | 4.00 |
| | JHOD | 20.73 | 0.32 | 3.87 | 2.52 | 4.87 |
| ARCTIC | GrabNet-Refine | 37.23 | 1.22 | 14.20 | 12.17 | 1.33 |
| | JHOD | 23.92 | 0.42 | 4.55 | 2.91 | 4.13 |
| Self-generated | GrabNet-Refine | - | 1.17 | 13.82 | 10.03 | 1.33 |
| | JHOD(uncond.) | - | 0.51 | 6.71 | 7.13 | 3.53 |

Table 2: The ablation study about the impact of training data on the generation quality over unseen objects from ARCTIC (Fan et al., 2023). Adding more training data consistently boosts the generation quality on unseen objects.

| OakInk-Shape | GRAB | Object-only Data | FID ↓ | Pene. Dep. ↓ | Intsec. Vol. ↓ | Sim. Disp. Mean ↓ | User Scor.↑ |
|---|---|---|---|---|---|---|---|
| ✓ | | | 27.29 | 0.51 | 4.89 | 3.22 | 3.87 |
| ✓ | ✓ | | 25.31 | 0.47 | 4.61 | 3.08 | 4.00 |
| ✓ | ✓ | ✓ | 23.92 | 0.42 | 4.55 | 2.91 | 4.13 |

**Sampling.** We follow the canonical progressive denoising process for diffusion models in the sampling stage. To sample for joint (unconditional) HOI generation, we synchronize the schedulers $t_H = t_O = t$. Similar to tevet2022human, at each step $t$, we predict a clean sample by $\hat{z}_0 = G_\Phi(z_t, t_H, t_O)$ from the corrupted code $z_t$ and then noise it back to $z_{t-1}$. During sampling for object-conditioned grasp generation, we keep $t_O = 0$ and $z_{t_H} = z_{t_H}^H \oplus z_0^O$. Then the denoiser predicts $\hat{z}_0 = G_\Phi(z_{t_H}, t_H)$ and then noise it back to $z_{t-1}$. In either the unconditional or the conditional generation, we repeat the process above along $t = T \longrightarrow 1$ until the final $\hat{z}_0$ is achieved after $T$ iterations.

## 4 Experiments

### 4.1 Setups

**Datasets.** We combine the data from multiple resources to train the model. GRAB (Taheri et al., 2020) contains human full-body poses together with 3D objects. We extract the hands from the full-body annotations. For OakInk (Yang et al., 2022), we use the official training split for training. We also use the contact-adapted synthetic grasp from the OakInk-Shape dataset for training. Besides the hand-object interaction data, we also leverage the rich resources of 3D object data to help train the object part in our model. The object data is also used to fine-tune the LION prior distribution. Fine-tuning prevents improper object shapes from the original distribution (pre-trained from ShapeNet (Chang et al., 2015)) such as sofas, chairs, and bookcases. We combine the objects from GRAB, OakInk (both the objects from OakInk-Image with grasp annotation and the objects from OakInk-shape without grasps), Affordpose (Ye et al., 2023b) and DexGraspNet (Wang et al., 2023) as the object data resource. We also leverage the DeepSDF (Park et al., 2019) as trained in Ink-base (Yang et al., 2022) to provide synthetic object data and include them in OakInk-Shape.

**Data Pre-Processing.** We follow the convention in OakInk-Shape to transform the hands into the object-centric frame. For data from GRAB (Taheri et al., 2020), we extract the MANO parameters from its original SMPL-X (Pavlakos et al., 2019) annotations. Then, we filter out the frames where the right-hand mesh has no contact with the object in training. For the object representation, we randomly sample 2048 points from mesh vertices to represent each object every time we draw the object in both training and inference to avoid overfitting a fixed set of point clouds. For the objects in the OakInk-Shape dataset, we sample the point clouds from the object mesh surfaces uniformly instead of sampling from the vertices because the annotated mesh vertices are too sparse on smooth surfaces.

Table 3: The ablation study about the impact of training data on the generation quality for joint generation of hand and object. Adding more data, either hand-obejct grasp data or object-only data, improves the generation quality.

| OakInk-Shape | GRAB | Object-only Data ↓ | Pene. Dep. ↓ | Intsec. Vol. ↓ | Sim. Disp. Mean ↓ | User Scor.↑ |
|:---:|:---:|:---:|:---:|:---:|:---:|:---:|
| ✓ | | | 0.92 | 8.12 | 8.23 | 3.33 |
| ✓ | ✓ | | 0.51 | 6.71 | 7.13 | 3.53 |
| ✓ | ✓ | ✓ | 0.43 | 5.55 | 6.98 | 3.73 |

Table 4: Ablation about asynchronous denoising schedulers. We evaluate on the generated object shapes by JHOD.

| Asyn. | Pene. Dep. ↓ | Intsec. Vol. ↓ | Sim. Disp. Mean ↓ |
|:---:|:---:|:---:|:---:|
| | 0.57 | 6.52 | 7.88 |
| ✓ | 0.43 | 5.55 | 6.98 |

**Implementation.** The DeepSDF in Ink (Yang et al., 2022) is trained per category to derive better details and fidelity when interpolating object shapes. We use the DeepSDF to provide synthetic object shapes. For the encoder network to transform modality features to latent codes, we always use 2-layer MLP networks with a hidden dimension of 1024 and an output dimension of 512. For the denoiser, we follow MDM (Tevet et al., 2022) to use a transformer encoder-only backbone-based diffusion network. After decoding the object latent code by the LION decoder, we derive the point cloud set. For visualization purposes, we conduct surface reconstruction from the generated object point cloud. Though advanced surface reconstruction techniques such as some learned solver (Peng et al., 2021) can provide more details, this part is not our focus and we wish to keep consistency for out-of-domain objects generated. Therefore, we choose the classic Alpha Shape algorithm (Edelsbrunner et al., 1983). Without category labels or other constraints, it is challenging to generate object shapes with highly carved details and it is not our focus in this paper.

**Baseline Methods** There is no commonly adopted benchmark in the area of grasp generation and some similar methods can follow different evaluation protocols. For example, the provided evaluation protocol of G-HOP (Ye et al., 2024) is for grasp reconstruction instead of generation and text description or object label is necessary for its generation mode which is not required in our proposed method. On the other hand, another line of works, such as UGG (Lu et al., 2023) and DexDiffuer (Weng et al., 2024) focus on grasp generation but within the domain of robotic dexterous hand thus there is no trivial way to compare with their in the same setting. In this work, we focus on hand grasp generation with only object shape as condition or without any condition. We select the widely adopted method GrabNet (Taheri et al., 2020) as the baseline method to compare with in this section. We also use its refined version GrabNet-Refine in some certain comparisons.

**Evaluation and Metrics.** Unlike the reconstruction tasks where the ground truth is available, it is difficult to do the quantitative evaluation for generation models. We hold the objects from the OakInk-Shape test set and ARCTIC (Fan et al., 2023) dataset for quantitative evaluations. We measure the quality of generated grasp by **FID (Frechet Inception Distance)** (Heusel et al., 2017) between the images rendered from the ground truth grasps and the generated grasps. We note that we can not directly calculate the FID score on generated 3D parameters as there is no commonly used encoder for this purpose and implementing it ourselves can cause many ambiguities. Therefore, we follow the practice Chan et al. (2022); Gao et al. (2022) of computing metrics on rendered images. Also, as the variance of camera views can significantly impact the image distribution, we render three views of each GT or generated grasp. We render three views because penetration or implausible contact may only be visible from certain angles. This multi-view rendering can provide more comprehensive qualitative evaluation but also reduces the potential bias introduced by the setup of single camera view. We also measure the model performance by directly checking the 3D asset quality. We follow previous studies (Yang et al., 2022; 2021) to use three metrics: (1) **the penetration depth (Pene. Depth)** between hand and object mesh, (2) **the solid intersection volume (Intsec. Vol.)** between them and (3) **the mean displacement in simulation** following (Hasson et al., 2019). Finally, we perform a user study to measure the plausibility and the truthfulness of the generated grasps

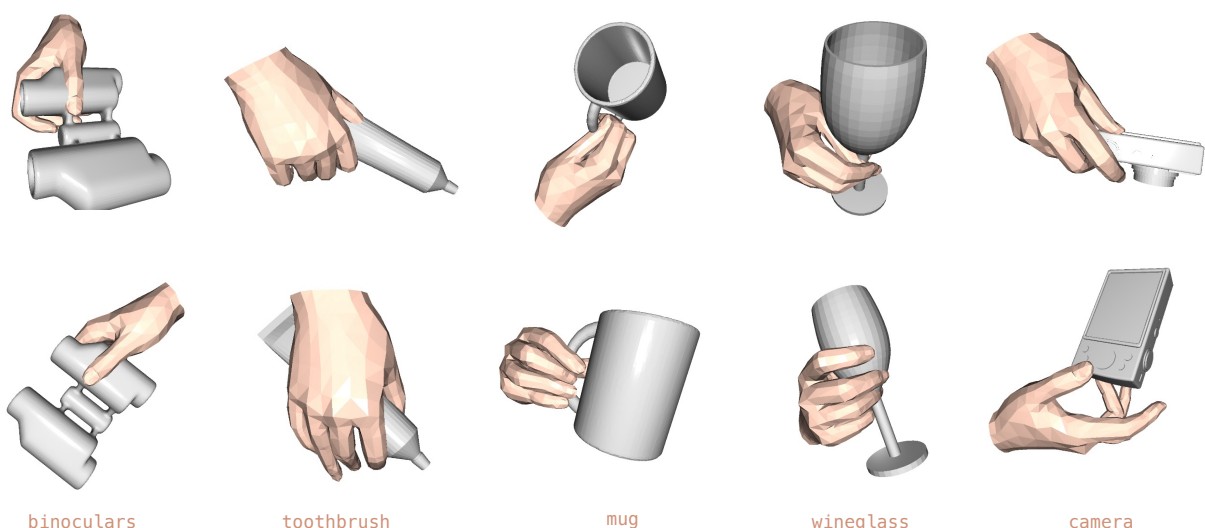

| binoculars | toothbrush | mug | wineglass | camera |

Figure 3: Generated samples on objects from GRAB (Taheri et al., 2020) dataset.

by scoring (1-5). We invite 15 participants to rate the quality of batches of the mixture of 10 rendering of ground truth grasps and 10 generated grasps. The score 1 indicates "totally fake and implausible" and 5 indicates "plausible enough to be real". Each participant evaluates by averaging the scores over 10 randomly selected batches.

## 4.2 Object-Conditioned Grasp Generation

Our method can generate visually plausible grasp configurations even though the model has not seen these objects during training. We generate samples on the objects from GRAB and OakInk test sets as shown in Figure 3 and Figure 4 . We perform the quantitative evaluation on the objects unseen during training with the previously introduced metrics. We follow the previous practice (Yang et al., 2022) to use the widely adopted GrabNet (Taheri et al., 2020) and its refined version GrabNet-refine (Yang et al., 2022) as the baseline models. We train the models on the GRAB and OakInk-shape train splits. The results are shown in Table 1.

Compared to the baseline methods, our proposed method achieves better generation quality per four metrics: FID, Mesh Penetration Depth, Intersection Volume, and Mean Simulation Displacement. The performance advantages are demonstrated on objects from the OakInk test set, generated by our method or ARCTIC dataset.

OakInk training set and training set contain similar objects though not the same. Such a biased similarity between the training and test sets exists in many HOI datasets and conceals the limitation of generating grasps on unseen objects. So we also test on the ARCTIC dataset, which is more confidently out-of-distribution from training. GrabNet-Refine fails to generate grasps with decent quality while the advantage of our method becomes more significant. GrabNet-Refine's generation quality is inferior to our method with a much larger margin by all metrics. The experiment reveals that some existing methods, such as GrabNet, face difficulty in generalizing to object shapes that are significantly different from training data. On the other hand, our method learns more generalizable and diverse object prior information and more universal grasp generation ability.

## 4.3 Unconditional Grasp Generation

Our proposed method learns the joint distribution of the latent representation of hand and object. It leverages large-scale object datasets to learn a sufficiently generalizable object shape embedding. To measure the quality of unconditional hand-object grasp, the results are shown in the last row in Table 1. The metric

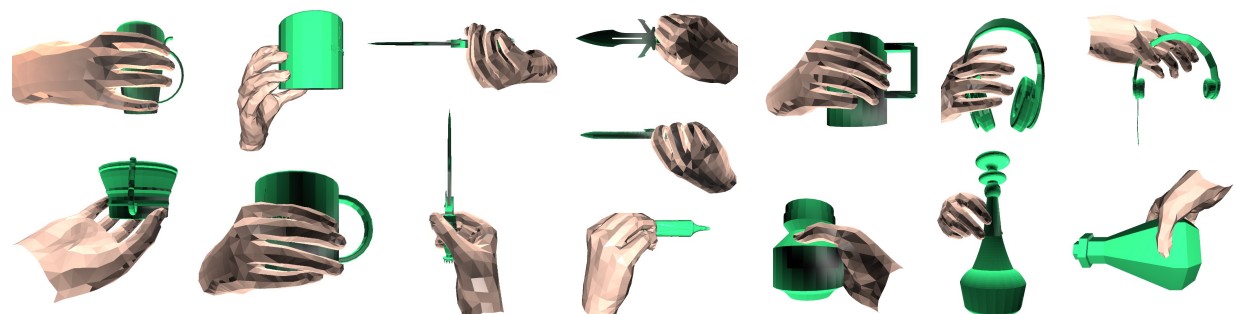

Figure 4: Generated samples on objects from OakInk (Yang et al., 2022) dataset.

scores indicate that the unconditional hand grasp generation quality is also good and close to the generation quality on ARCTIC objects. Compared to the object-conditioned generated grasps from JHOD, the quality of unconditional generation is inferior, which is in fact as expected. Here, we also provide the evaluation results by GrabNet-Refine on the same set of objects generated by our method for reference. And GrabNet-Refine fails to generalize to this set of unseen objects again just as its failure on ARCTIC objects. Such failure of generalizability is previously under-explored because the training/test splits of a HOI dataset usually contain objects from certain categories, with similar scale, geometry, and affordance. The method trained on the training split can learn significantly biased in-domain knowledge about the objects in the test splits. To generate hand grasps on more out-of-domain object shapes would be a challenge to all existing methods.

By designing the method capable of training with object-only data and disentangling the modality representations in the latent diffusion, our method not only has the advantage of generating the object and the hand grasp simultaneously but also achieves much better robustness to generate grasps on out-of-domain objects.

### 4.4 Ablation Study

We now provide ablation studies to show the contributions of different resources to our method's final performance.

**Ablation of grasp generation on unseen objects.** To provide transparent experimental conclusions, we ablate the training data on the model generation quality. We use the objects from ARCTIC (Fan et al., 2023) to measure the conditional generation quality on unseen objects. The quantitative evaluation results are shown in Table 2. Compared to using only OakInk-Shape data for training, adding GRAB data and object shapes from Object-only datasets both improves the generalization qualities. The improvement from extra GRAB data is easy to explain as training with more HOI data improves the generalizability of the model while it is interesting that adding the object-only datasets also boosts the performance. We believe this is because the extra object data improves the robustness and generalizability of the object encoding. It extends the latent representation expressiveness that the diffusion model learns to generate.

**Ablation of unconditional generation quality.** The ability to generate both hand and object to form a grasp is another main contribution. We are interested in whether it can benefit from the additional training data as well. We present the ablation study in Table 3. Similar to the conditional generation ablation study results, the unconditional generation quality also benefits from additional training data. The metrics of Pene. Dep., Intsec. Vol. and Sim. Disp. Mean indicate that the generated hand-object pair improves plausibility along with adding more training data. The increasing user scoring result suggests that the generated object and hand also become more and more visually realistic along with adding more training data.

**Ablation of synchronous denoising schedulers.** As one of the main implementation innovations of JHOD, we conduct an ablation about Asynchronous Denoising Schedulers in Table 4. For the fairness of the comparison, we keep the curriculum of the training the same by mixing data samples from different

resources. Intuitively, we designed this module to decouple the noise level in hand latent and object latent so that the model could better use the data with heterogeneous annotations and learn to denoise a certain modality. Without Asynchronous Denoising Schedulers, the noise diffusion and denoising time step of the object part is the same as the hand latent code. It makes the training biased to the HOI dataset where both modalities are available for supervision. Adding Asynchronous Denoising Schedulers, the denoising is learned with more independence for the object part and the hand part. The model can better learn object shape coding to generate hand grasps. Therefore, when evaluating on the out-of-domain objects generated by JHOD, the generation quality is improved as expected.

## 5   Conclusion

In this work, we propose a joint diffusion model for generating hand-object grasps. By encoding the modalities of objects and hands into a unified latent space, our model can generate grasps both jointly and conditionally. We address the limitations of existing methods that rely on limited hand-object interaction (HOI) data for training and explore strategies for learning grasp generation that generalizes to a wider variety of objects. Our approach leverages data with partial annotations, thereby mitigating the dependency on fully annotated datasets. The generated grasps, in both unconditional and conditional settings, demonstrate high quality. We believe that our method offers a significant advancement in learning generalizable and robust hand-object grasp generation, even with limited fully annotated data.

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

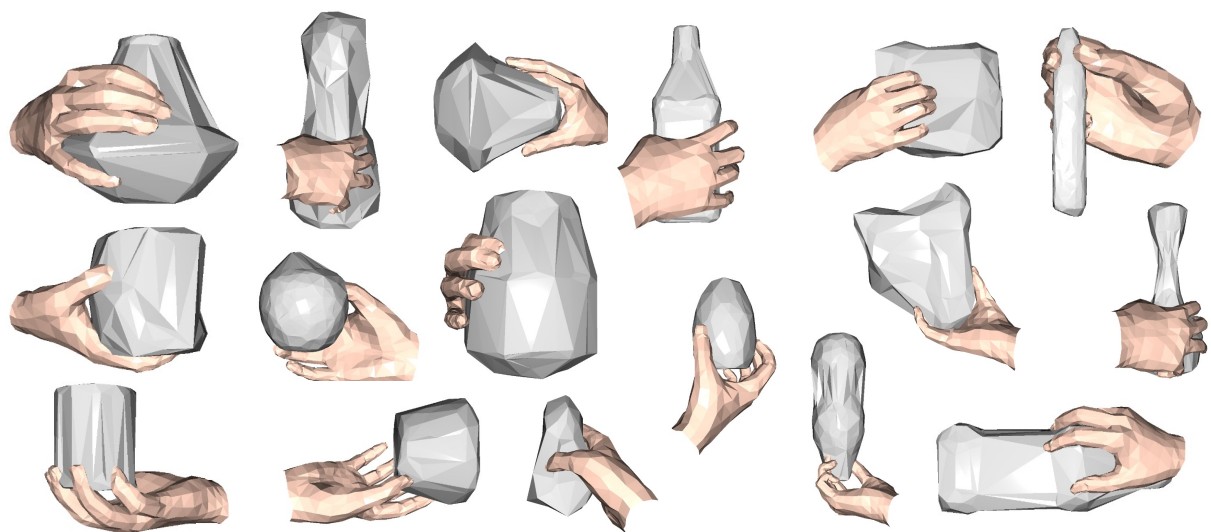

Figure 5: Samples of the unconditional generation of hand grasp together with objects.

## A  Grasp Diversity

Given a single object shape, we could generate a set of different grasps onto it. Though diversity is expected to be constrained by the hand grasp data available in training, there are some other implicit but more fundamental cues to help generate diverse hand grasp. This underlying but fundamental constraint can be leveraged to enhance the generation robustness over unseen objects if the feature of the object shape is sufficiently generalizable. For example, the model can learn to avoid penetration between the object volume and the hand shape and certain fingers should be close to the surface of objects to form a physically valid grasp. We generate grasps with unseen objects from the OakInk-shape test set as the condition in Figure 6. We observe some grasp patterns not provided in the training set, for example, holding a dagger between two fingers. As the implicit concept of forming a valid grasp is always conditioned to object shapes, our strategy of exposing the model training to more diverse object shapes can help to learn grasp patterns on unseen and even out-of-domain objects. We believe this is a key reason for making our method outstanding when generating grasps on out-of-domain objects.

## B  Dataset Statistics

Here we provide more details about the statistics of the datasets involved in our training and evaluation in Table 5. The HOI datasets with deformable and articulable hand models, i.e., MANO (Romero et al., 2022), face severe limitations of object resources. Combining the training set of GRAB (Taheri et al., 2020) and OakInk-Image (Yang et al., 2022) datasets still make just ∼100 objects. On the other hand, Afford-Pose (Jian et al., 2023) and DexGraspNet (Wang et al., 2023) contain more object shapes but the different choices of hand models make them hard to integrate with MANO-parameterized datasets. Fortunately, we have the large-scale 3D object shape dataset ShapeNet (Chang et al., 2015) with more than 50,000 objects. We combine the objects from these datasets in the training for the object part. They help to construct a more universal prior distribution for object generation and the grasp posterior distribution.

## C  Grasp-conditioned Image Generation

We demonstrate using JHOD as a creativity tool to generate 3D grasps and the corresponding images. In Figure 7, we first use JHOD to sample interaction pairs of 3D hands and objects. With the rendering of the 3D hand-object grasp as the input, we applied Adobe Firefly (Adobe, 2024), an existing tool to generate the images with the depth and edge reference of the 3D pairs. The hand pose and the geometry

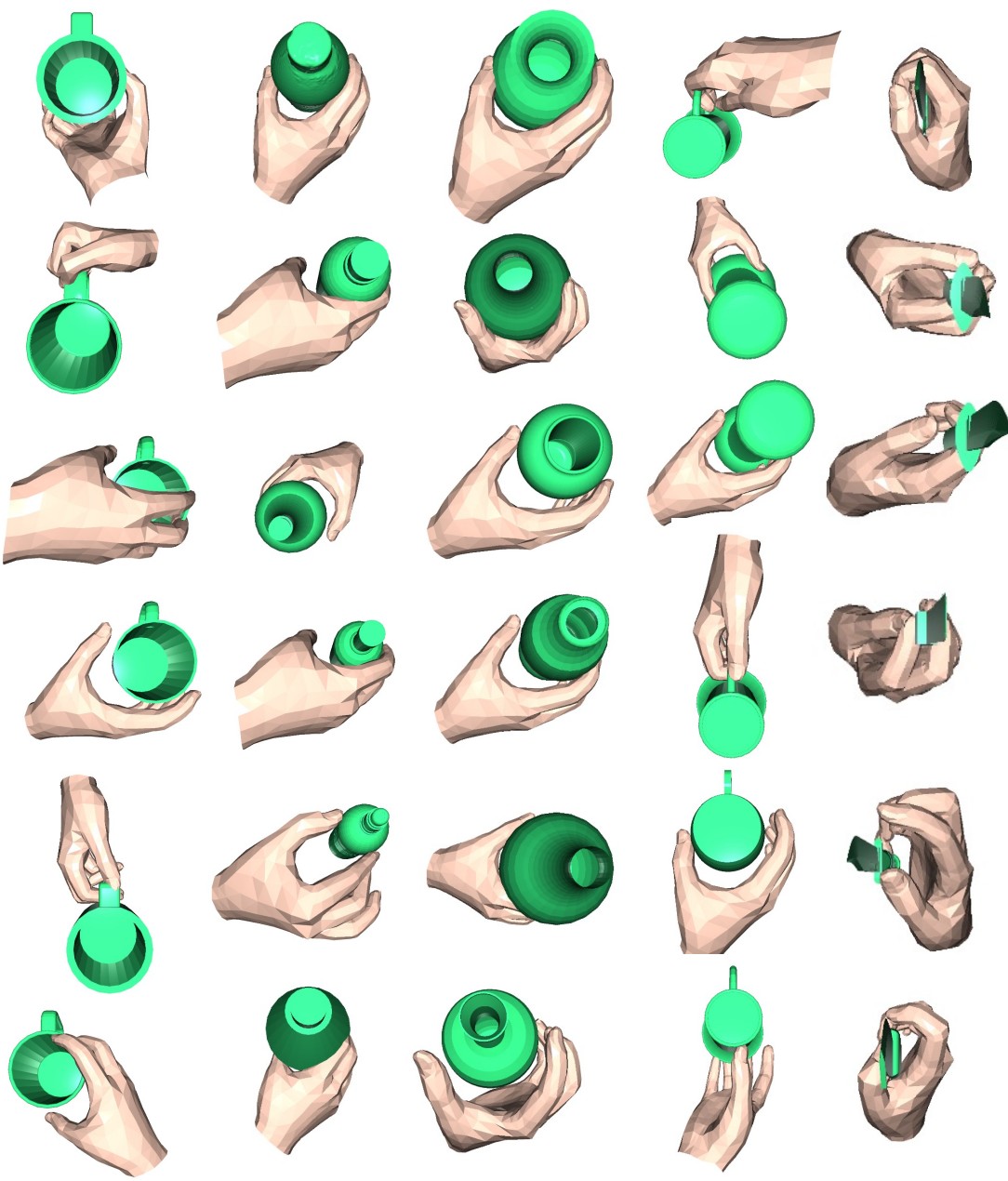

Figure 6: Given a single object, our method is capable of generating different grasps on it.

Table 5: Statistics of the datasets used for training. ShapeNet contains the most object assets. However it is a generic object shape dataset, thus many included objects are not proper for hand grasp. OakInk-Shape contains rich grasp and object data but can not provide annotation for object transformation. We use the grasp data from GRAB and the object data from AffordPose as the supplement during training.

| Datasets | #obj | #grasp | real/syn. | hand model |
|---|---|---|---|---|
| ShapeNet | 51,300 | - | real | - |
| GRAB (Taheri et al., 2020) | 51 | 1.3k | real | MANO (Romero et al., 2022) |
| OakInk-Image (Yang et al., 2022) | 100 | 49k | real | MANO (Romero et al., 2022) |
| OakInk-Shape (Yang et al., 2022) | 1,700 | - | real + synthetic | MANO (Romero et al., 2022) |
| AffordPose (Jian et al., 2023) | 641 | 26k | synthetic | GraspIt (Miller & Allen, 2004) |
| DexGraspNet (Wang et al., 2023) | 5,355 | 1.32M | synthetic | ShadowHand (sha) |

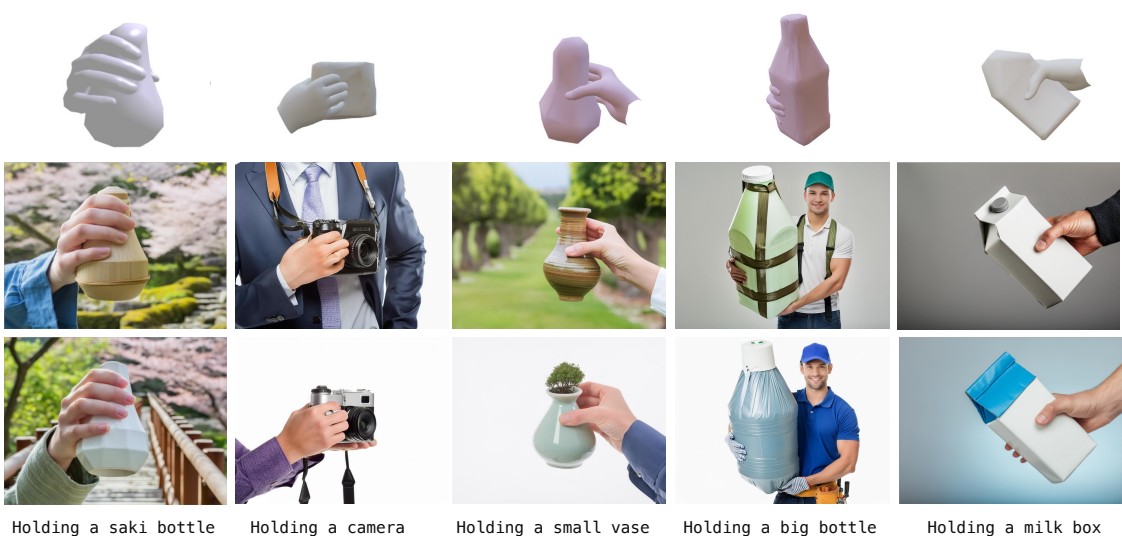

Holding a saki bottle    Holding a camera    Holding a small vase    Holding a big bottle    Holding a milk box

Figure 7: Samples of images generated by Adobe Firefly with synthesized hand grasp by our proposed method as the condition.

provide guidance for Firefly to complement the details given simple text prompts. In Figure 8, compared to grasping images authorized purely by text prompts, images guided by JHOD's output have more consistent gestures, better-aligned object geometry and boundaries, and fewer hand artifacts. Therefore, with the rise of text-to-image generation, our hand-object grasp generation model can serve as a proxy to enhance the consistency among multiple instances and the visual plausibilities. We also provide a closer look at their detailed in Figure 9 to compare the results with and without the generated grasp as a proxy. At each row, the images are generated from the same text prompt as shown at the bottom. There are two main benefits of generating images conditioned to the grasp. It first allows a set of images with aligned hand and object geometry and boundaries which can be useful for image and video editing. On the other hand, even though the Adobe Firefly generator has shown a significant advantage over the public Stable Diffusion in eliminating artifacts, we still observed many finger artifacts when generating the images without a grasp condition. Fortunately, with the generated grasp, including the object shape and the hand, as the condition, the image generator can produce significantly fewer artifacts, especially artifact fingers, which have been a notorious issue in image generation recently. We provided some zoomed-in examples at the bottom. Moreover, compared to previous works using hand shapes to guide image generation (Ye et al., 2023b; Narasimhaswamy et al., 2024), we could generate both the object and the hand. Therefore, we could use the whole scene of hand-object interaction as the condition and thus enable more controllable details in the generated image.

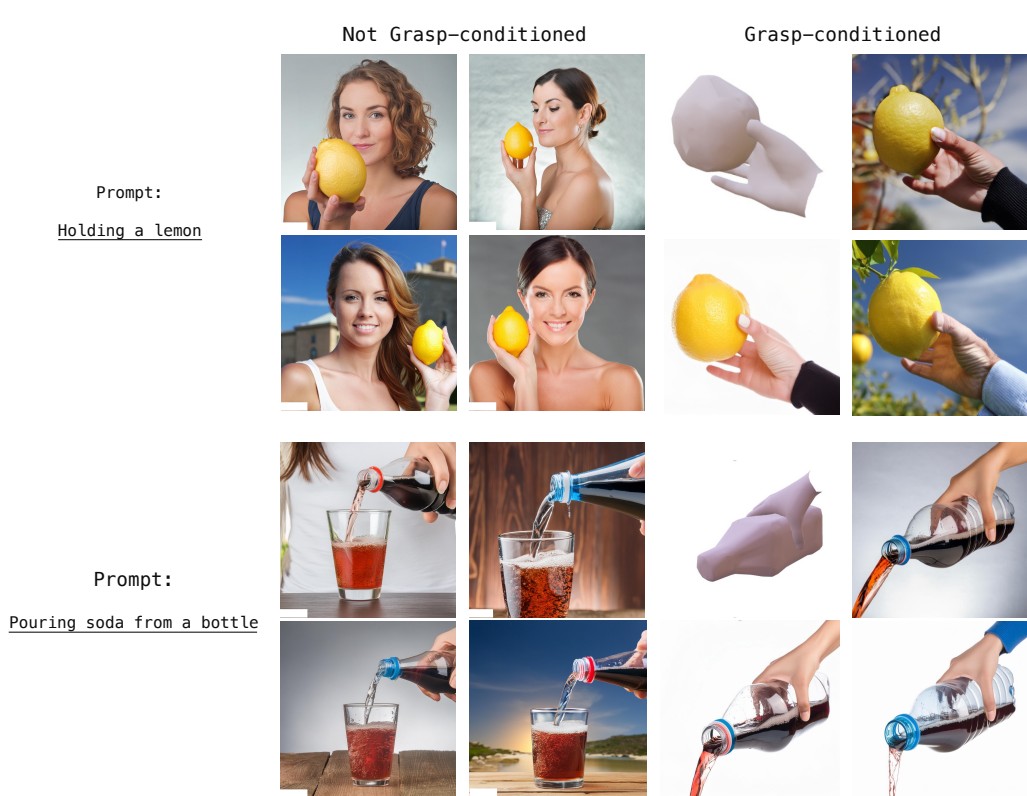

Figure 8: Using the same text-prompted image generation tool, we synthesize photo-realistic images with and without the grasp as the geometry condition. The grasp-conditioned image generation can have broad applications in image and video editing.

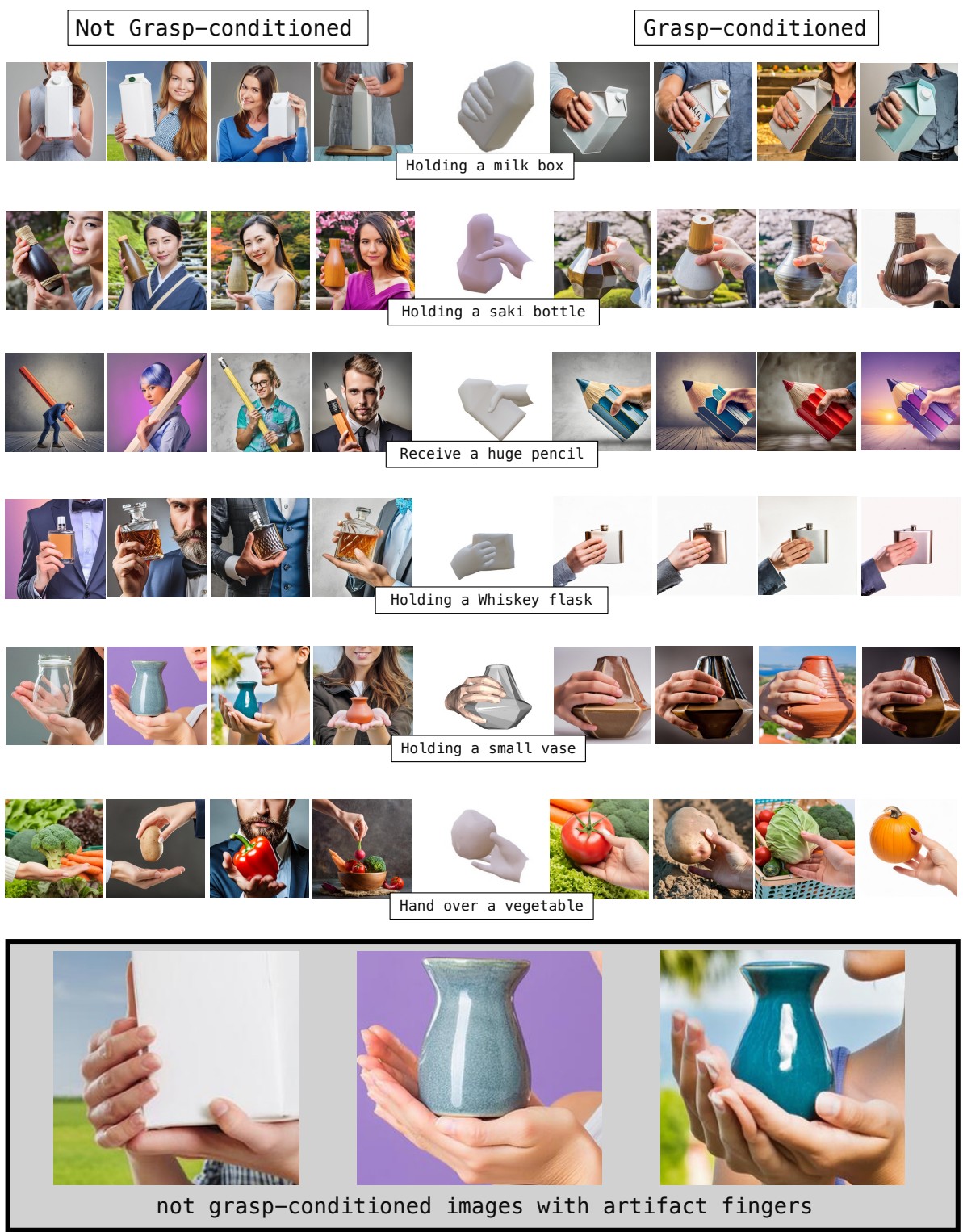

Figure 9: More examples of generating images of grasps with and without the condition from our generated grasps by the same image generation tool. Without a plausible grasp as the condition, there are more frequent unrealistic artifacts in the generated images, especially the number and pose of fingers. We provide some zoomed-in bad examples at the bottom.

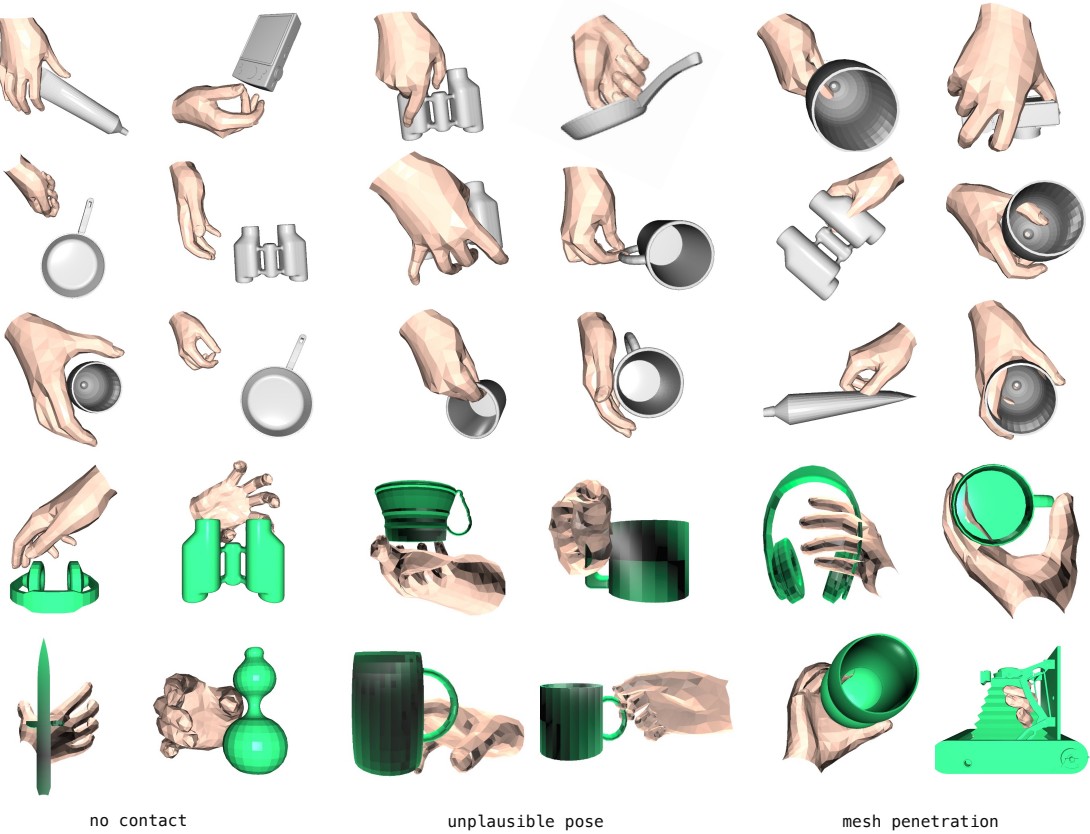

no contact          unplausible pose          mesh penetration

Figure 10: Bad samples of conditional grasp generation given objects from GRAB (gray) and OakInk-Shape (green).

# D    Failure cases

To provide a transparent evaluation of our proposed method, we still notice some failure cases on the unseen objects from OakInk and GRAB datasets as shown in Figure 10. There are in general three patterns of failures: (1) the hand and the object have no contact, making the grasp physically implausible; (2) the hand skin is twisted or the pose is not able to grasp the object in human common sense; (3) the mesh of object and hand has penetration and intersection. There can be some reasons that make these happen. First of all, we do not explicitly supervise the contact between the object surface and hand as this is a rare annotation on many data sources, and calculating it can be computationally expensive. On the other hand, some implausible grasps have pretty good visual quality as shown in the middle two columns in the figure. However, according to our life experience, we know that the grasp is not physically plausible to manipulate the object. This has gone beyond our scope in this work as we do not have any physics-aware supervision such as the physical demonstration in a physics simulator. Finally, we use the object point cloud to represent object shapes to allow grasp generation on universal objects without a template, this causes the potential penetration between object mesh and hand mesh as the object mesh is ambiguous and unknown to our method. Despite the failure cases, we note that they make only a very small portion of the generated samples (less than 10% by a rough estimation). We show them here for transparency and to help discussion about future works in this area.

# E  Hand Contact with Object

Hand contact with the object is a key component of a valid hand-object grasp. In Figure 11, we visualize the contact area on the hand mesh when the hand is posed to grasp an object. We provide the visualization of the hand-object grasp in three different camera views to showcase the full configuration of the grasp, and also a hand-only visualization to more clearly justify the contact area (in red). Following existing practice while making a stricter standard to justify a valid "contact", we set the distance threshold between the hand mesh face and the object mesh face to be 0.005 meter for the visualization.

# F  Qualitative Comparison

In addition to the quantitative results presented in the experiment sections, we now provide qualitative comparisons between our method and the baseline GrabNet (Taheri et al., 2020), as shown in Figure 12. We randomly sample objects from the "bottle" category in OakInk that were not seen during training. Grasps are generated for each object without any cherry-picking. In each result pair, the left-hand grasp is produced by our method, while the right-hand grasp is from GrabNet. As shown, GrabNet generates grasps that penetrate the object in pairs (1), (2), (4), and (6), and lacks plausible contact in pairs (3) and (8). In contrast, our method produces more realistic grasps in most cases, although some penetration is still observed in pairs (2) and (6). Overall, these comparisons suggest that our method achieves superior qualitative performance in grasp generation.

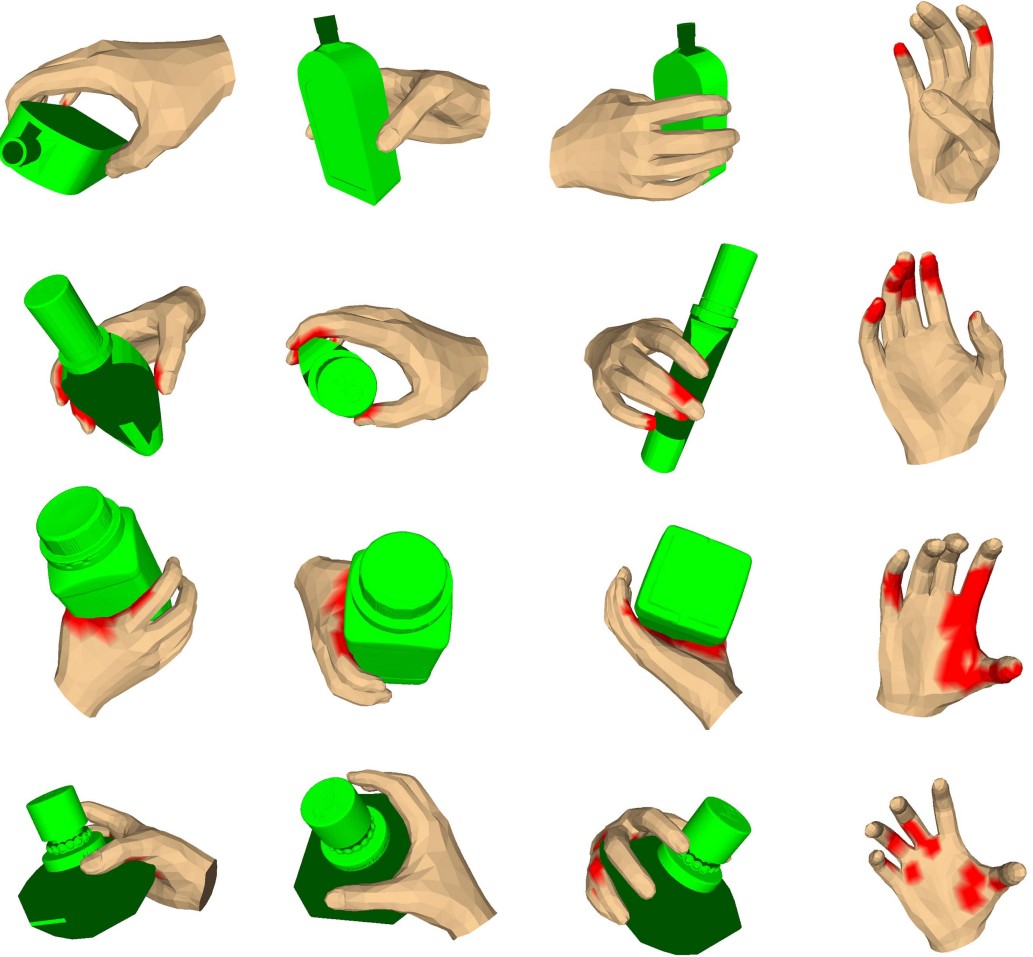

Figure 11: Visualization of the generated hand-object grasp and the corresponding hand contact area. Here, we set the distance threshold to justify the contact between a hand mesh face and an object mesh face to be 0.005 meters. The contact area is rendered in red.

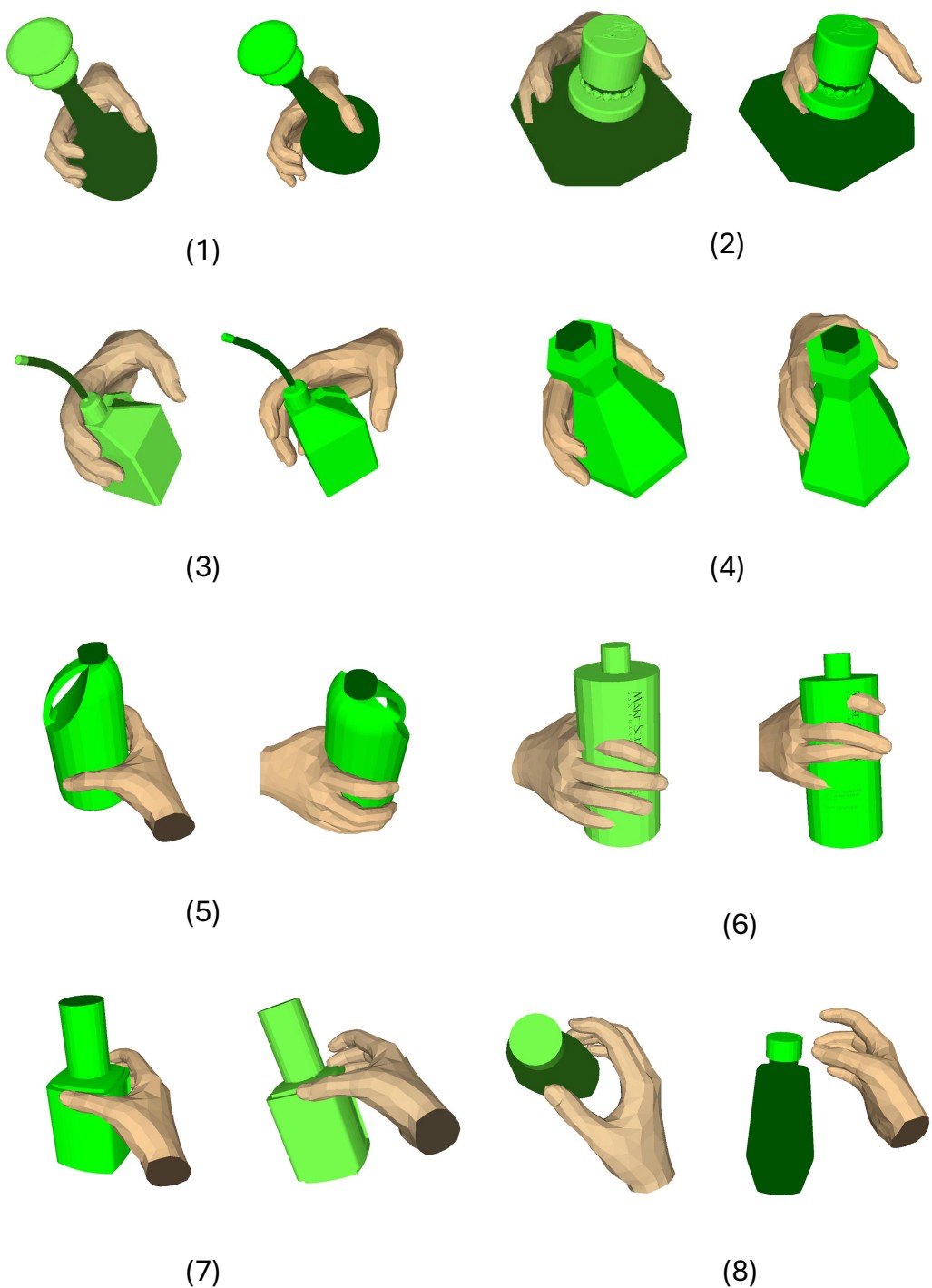

Figure 12: Grasps generated by our proposed method and GrabNet (Taheri et al., 2020). In each pair of grasps, the left-hand grasp is generated by our method and the right-hand one is generated by GrabNet.

