# OpenReview forum: "Joint Diffusion for Universal Hand-Object Grasp Generation"
_TMLR — Accepted by TMLR_

### Review · Reviewer_TZMo · 2025-06-28

**Summary Of Contributions:**

This paper introduces Joint Hand-Object Diffusion (JHOD), a novel latent diffusion model for generating hand-object grasp configurations in both unconditional and object-conditioned settings. Unlike previous works that rely on limited hand-object interaction datasets, JHOD leverages large-scale object-only datasets to learn more robust object embeddings and jointly models hand and object representations in a unified latent space. A key technical contribution is the use of asynchronous denoising schedulers that allow training on heterogeneous data (e.g., object-only or grasp-only), which boosts generalization and flexibility. The authors conduct thorough evaluations and ablations, showing strong generalization to unseen and out-of-domain objects, outperforming GrabNet and GrabNet-Refine by a notable margin. Additionally, the model enables creative applications such as grasp-conditioned photorealistic image generation.

**Audience:**

Yes

**Claims And Evidence:**

Yes

**Requested Changes:**

1. Could you elaborate on the mechanism by which training the object reconstruction (via diffusion on object-only data) helps improve the grasp generation quality? Is it purely through improved object embeddings or is there implicit transfer to hand latent decoding?

2. In the training stage with object-only data, the hand part of the joint latent code is sampled from noise. How does the model avoid learning a distribution that conflicts with the objectives of conditional/unconditional grasp generation, where the hand code is expected to be meaningful?

**Strengths And Weaknesses:**

**Strengths:**

1. Unified Latent Diffusion Framework: The proposed model can handle both unconditional and conditional generation, which is a significant step forward in general-purpose hand-object interaction modeling.

2. Extensive Experiments: The paper evaluates on multiple datasets (GRAB, OakInk, ARCTIC), includes detailed ablations (data sources, scheduler design), and uses a range of metrics (FID, penetration depth, intersection volume, user study scores).

3. Generality & Practical Utility: The method supports multiple downstream applications such as photorealistic image synthesis, without requiring language prompts or category labels.

**Weaknesses:**

1. Motivation for Using Object-Only Data Is Underexplained: While the idea of leveraging large-scale object-only datasets to improve hand-object grasp generation is compelling and shows empirical gains (e.g., in Table 2 and 3), the paper lacks a clear theoretical or intuitive justification for why object shape reconstruction via diffusion contributes to grasp generation, especially in the absence of grasp-specific supervision.

2. Joint Latent Code Denoising Ambiguity: The proposed asynchronous denoising schedule enables training on heterogeneous data, including object-only samples by filling the hand latent with Gaussian noise. However, it is unclear whether this training setup introduces distributional mismatch or learning conflicts between object-only generation and unconditional/conditional generation.

---

> ### Author Response · Authors · 2025-07-28
>
> We appreciate the comments from the reviewer TZMo. Regarding the questions from the reviewer, we have some clarification and feedback:
>
> 1. We clarify that the object-only training loss (Eq. 13) is designed to improve the object latent code by exposing it to a broader distribution of object shapes. The motivation behind this design is that HOI datasets typically contain a limited number of object categories, which is insufficient for learning a generalizable object reconstruction model. As a result, the object latent codes learned solely from HOI data tend to overfit and fail to generalize to unseen objects, leading to poor grasp generation performance.
> To address this, we incorporate object-only training using large-scale object datasets, which are known to support better generalization across diverse object geometries. By optimizing Eq. 13, we refine the object branch of the joint latent space, making it more robust to novel object shapes at inference time.
>
> 2. Importantly, Eq. 13 does not apply any supervision or gradients to the hand branch of the latent space. While a hand code is technically present (as part of the joint latent vector), it is randomly initialized and remains untrained during this phase, consistent with the reviewer’s observation. We do not decode the hand code into a pose during object-only training, and it does not influence the object reconstruction process. The hand code is only trained and supervised in the conditional generation (Eq. 12) and unconditional generation (Eq. 11) settings.

---

> > ### Comment · Reviewer_TZMo · 2025-08-12
> > **Response to Authors**
> >
> > Thank you for the detailed clarifications. After reading your response, I now have a clearer understanding of the model design. In particular, my earlier concerns about the motivation for object-only training and the potential conflict in the joint latent code are largely resolved.
> >
> > From my current interpretation, Eq. 9 suggests that the hand denoising process should be conditioned on the object latent code $z^O_{t_O}$, not completely independent. One example supporting this is in the object-conditioned grasp generation setting, where $t_O = 0$. Meanwhile, unconditional generation in the paper also implies such conditioning. Since the object denoising process is a standalone reconstruction, it allows partial training with object-only data, and the gains shown in Tables 2 and 3 confirm the effectiveness of this design. If my interpretation is correct, then my initial concerns are addressed. However, I recommend improving the writing and presentation—especially around Eq. 9 and the asynchronous denoising scheduler—to avoid ambiguity for future readers.
> >
> > I also appreciate the discussion on evaluation metrics. The newly added visualizations are particularly helpful for understanding the results.
> >
> > Lastly, I noticed that the citation for the ARCTIC dataset seems to point to a different paper. You may want to double-check and correct this reference.

---

> > > ### Author Response · Authors · 2025-08-23
> > >
> > > We appreciate the reviewer's confirming that some concerns are addressed and we will improve the draft further referring to the suggestions.
> > >
> > > Yes, as the model is learning a joint distribution for hands and objects to form a valid grasp. Therefore, in the canonical settings, the denoising of hands should be related to objects. Eq 9 is a direct rewriting from Eq 3 to emphasize that two different schedulers $t_O$ and $t_H$ are used for objects and hands separately instead of a single and shared one. The current equations make some confusion here and we will improve the writing.
> > >
> > > I double checked the citation entry and seems that *Fan et. al. 2023* is the correct reference entry for ARCTIC dataset. We are keeping improving the writing and revising the paper.
> > >
> > >
> > > **Reference**:
> > >
> > >
> > > Zicong Fan, Omid Taheri, Dimitrios Tzionas, Muhammed Kocabas, Manuel Kaufmann, Michael J Black,
> > > and Otmar Hilliges. Arctic: A dataset for dexterous bimanual hand-object manipulation. In Proceedings
> > > of the IEEE/CVF Conference on Computer Vision and Pattern Recognition, pp. 12943–12954, 2023.

---

### Review · Reviewer_vwUU · 2025-06-30

**Summary Of Contributions:**

This paper introduces a diffusion model for joint Hand-Object grasp generation (both conditional, unconditional). The proposed Joint Hand-Object Diffusion (JHOD) model is trained with the combination of 1) existing hand-object interaction datasets, 2) object shape datasets, and 3) synthetic object datasets, and the proposed diffusion training scheme is designed to handle such heterogeneous supervision from partial annotations. Given such a mixed dataset, the authors train the latent diffusion model (JHOD) denoiser network, hand encoder, and decoder jointly, while the object encoder and decoder are borrowed and fixed from the existing work, LION. The performance is evaluated using FID scores of mesh renderings, and some feasibility terms for checking hand-object penetration, and finally, the user preference scores.

**Audience:**

Yes

**Claims And Evidence:**

Yes

**Requested Changes:**

Overall, this reviewer found the paper interesting and sound. This reviewer expects the authors to address the concerns about the evaluation side, especially with 1) more analyses and justifications on using FID, 2) more qualitative samples, and 3) visualizing the contact or affordance of the generated hand-object grasp scenes.

**Strengths And Weaknesses:**

**Strengths**

- This paper introduces an interesting way to leverage the existing heterogeneous 3D datasets, including the hand-object grasp dataset, the object shape dataset, and the synthesized datasets.
- While overall design choices are fairly well known and simple, the idea of leveraging external datasets and model (e.g., LION) for overcoming the scarcity of the data in the field is interesting. Specifically, the authors' effort to avoid distinguishing object classes when training the diffusion model is interesting: they attempted to encode object shapes based solely on the objects' geometric cues. Although they used the pre-trained object encoder-decoder for this, this reviewer thinks that it's better than existing approaches, which typically rely only on limited hand-object grasp datasets.

**Weaknesses**
- This reviewer has concerns regarding the evaluation part.
- First, the authors propose to measure the FID score from the rendered images of the 3D hand and objects. While this reviewer agrees that there are no outstanding metrics to measure such a generation task, this reviewer thinks that the FID of rendered 3D objects is not so robust. Specifically, when measuring the feasibility or realism of the hand-object grasp, the camera views for rendering the 3D scene would be critical factors that may perturb the FID scores. How did the authors select the views for rendering the scene? More details, analyses, and justifications would be appreciated, e.g., providing some sample images to measure FID (both GT and generated sets), investigating the influence of the number of views or view angles on the FID score, etc.
- Next, the provided qualitative samples are limited. The samples presented in Figures 3, 4, and 5,6 are heavily overlapped. Moreover, it would be more convincing if the authors show more qualitative samples for the object-conditioned grasp generation on the unseen or unusual object shapes.
- It would be good if the authors could also visualize contact or affordance information for the generated hand-object grasp samples. While the authors report the quantitative results and the quantitative ablation study for it, this reviewer thinks it would be more compelling if the authors could visualize the contact or affordance, especially for the ablation study (Table 3). Since the metrics for penetration, intersection, etc, have improved when using more datasets (in Table 3), this reviewer thinks the qualitative results for contact points would also be improved.

---

> ### Author Response · Authors · 2025-07-28
>
> We sincerely appreciate Reviewer vwUU’s thoughtful comments. Below, we address your concerns and questions in detail:
>
> 1. FID Evaluation:
> Directly measuring distribution alignment between ground truth (GT) and predicted grasps in 3D space is challenging. Consequently, many 3D generation works, such as [1,2], render images from the generated 3D assets and then compute metrics like FID or R-Precision in the 2D image space. Our work follows this standard practice.
> We acknowledge that camera viewpoints can influence FID evaluation. To mitigate this, we render three views of each GT or generated grasp. We use Open3D for mesh placement and rendering (the same tool used for all images in the paper), with the camera placed at three positions:
> $(extent×1.5,0,0),(0,extent×1.5,0),(0,0,extent×1.5)$,
> where extent is the maximum extent of the object. Because our hand grasps are predicted in an object-centric coordinate system, the object is always centered in the rendering scene. Thus, for each GT and generated image pair, both the object placement and camera setup are identical.
> We deliberately render three views instead of a single view (as is common in prior work) because penetration or implausible contact may only be visible from certain angles. This multi-view rendering not only provides more comprehensive qualitative evaluation but also reduces the potential bias introduced by any single camera view in the FID calculation.
>
> 2. Qualitative Comparison with GrabNet:
> In Figure 12 of the revised paper, we include a new set of qualitative comparisons between our method and GrabNet. The objects are randomly sampled without any cherry-picking. We believe this offers a clearer and more objective sense of our method’s generation quality.
>
> 3. Contact Visualizations:
> In Figure 11 of the revised paper, we introduce new visualizations highlighting the contact area on the hand surface. These visualizations provide a clearer understanding of the interaction between the generated hand and the object during grasping.
>
> References:
> [1] “Efficient Geometry-aware 3D Generative Adversarial Networks,” CVPR 2022.
> [2] “GET3D: A Generative Model of High Quality 3D Textured Shapes Learned from Images,” NeurIPS 2022.

---

### Review · Reviewer_wCFg · 2025-07-14

**Summary Of Contributions:**

The authors aim to generate hand-object grasps by diffusion. The authors argue that prior datasets for this are sometimes limited in the class of objects, or have different models for the hand geometry, causing poorer generations on objects that are out of distribution.

To handle this, the authors design a joint denoising diffusion model operating in both object-space and hand-space. Each modality (object, hand) is encoded into a latent $z_O, z_H$, which is concatenated into a longer latent $z$. Denoising happens over this $z$, which allows the denoising to process the object and hand modalities together. The denoised latent is then decoded into a generated object and hand.

Such a model is used in two ways: to generate samples of ${O, H}$ unconditionally, or to generate hand samples $H$ conditioned on a fixed object ${O}$. The latter is what we are most interested in for downstream use cases.

The object encoder-decoder is a frozen model, from the LION paper (Vahdat et al. 2022), which was trained exclusively on 3d object generation from a large set of objects, so the paper exclusively trains the hand generation, using the narrow hand-object datasets and relying on generalization from LION to handle newer objects. To train this, the paper proposes an asynchronous denoising schedule, which applies two different noise schedules to the object embedding and the hand embedding. This is done because the model will be called either with a fully noised object and hand, or a defined object and noised hand, and the denoiser needs to handle both these cases.

**Audience:**

Yes

**Claims And Evidence:**

Yes

**Requested Changes:**

Please update the references to wrap them in (parentheses) rather than in-lining all the author names. Please include qualitative examples of generated grasps from baseline methods.

**Strengths And Weaknesses:**

In general, as someone who would use grasp-pose generation downstream in robotics in particular, I have lower trust in evaluation metrics that do not use these downstream use cases. As mentioned in Section 4.1, it can be difficulty to do quantitative evaluation of hand grasps when we do not have a single ground truth like we do for reconstruction losses. The measures of penetration depth, intersection volume, etc. are good, but the paper would be stronger if there were some qualitative comparisons between generations from JHOD and GrabNet. I believe all the images are only from JHOD and it would be nice to see commentary on general trends that differ betwee JHOD and GrabNet, with examples.

Aside from this, the way diffusion is done is well-founded and is what you would generally do when training a multimodal diffusion model. I believe the paper is solid enough.

---

> ### Author Response · Authors · 2025-07-28
>
> We appreciate the valuable feedback from Reviewer wCFg. In response, we have revised the paper and uploaded a new version with the following updates:
>
> 1. We have corrected the in-text citation format by placing references within parentheses.
>
> 2. We added additional visualizations in Figures 11 and 12. Specifically, Figure 11 now includes contact area visualizations on the hand surface to offer further insight into the plausibility of the generated grasps. Figure 12 presents a more direct, apple-to-apple comparison between the grasps generated by our method and those from the baseline method, GrabNet, on the same unseen objects. We believe these new visualizations provide clearer and more comprehensive qualitative evidence of our method’s effectiveness.

---

### Decision · Action_Editor_bhSx · 2025-10-18

**Recommendation:** Accept with minor revision

**Additional Comments:**

Please address the clarity issues raised by reviewer TZMo, as well as evaluation related questions from reviewer vwUU, wCFg.

**Audience:**

Yes

**Audience Explanation:**

The paper studies a niche but interesting problem of joint hand object generation. All reviewed agree that this is an interesting problem and is relevant to the generative modeling and robotics community.

**Claims And Evidence:**

Yes

**Claims Explanation:**

All reviewers acknowledge the soundness of the proposed method. The results are significant, although there are rooms for improvement in terms of the evaluation protocol, as per reviewer  vwUU, wCFg.

---

> ### Author Response · Authors · 2025-11-18
> **camera-ready draft submitted**
>
> We appreciate the suggestions and comments from the Action Editor and all reviewers. We have revised the draft to address the clarity issues raised by the reviewers and have updated the evaluation section based on our discussions with them. The camera-ready draft has been submitted.